# Structure of a heteropolymeric type 4 pilus from a monoderm bacterium

Robin Anger[1], Laetitia Pieulle[2], Meriam Shahin[3], Odile Valette[2], Hugo Le Guenno[4], Artemis Kosta [4], Vladimir Pelicic [2,3] ✉ & Rémi Fronzes [1] ✉

Type 4 pili (T4P) are important virulence factors, which belong to a super-family of nanomachines ubiquitous in prokaryotes, called type 4 filaments (T4F). T4F are defined as helical polymers of type 4 pilins. Recent advances in cryo-electron microscopy (cryo-EM) led to structures of several T4F, revealing that the long N-terminal α-helix (α1) – the trademark of pilins – packs in the centre of the filaments to form a hydrophobic core. In diderm bacteria – all available bacterial T4F structures are from diderm species – a portion of α1 is melted (unfolded). Here we report that this architecture is conserved in phylogenetically distant monoderm species by determining the structure of *Streptococcus sanguinis* T4P. Our 3.7 Å resolution cryo-EM structure of *S. sanguinis* heteropolymeric T4P and the resulting full atomic model including all minor pilins highlight universal features of bacterial T4F and have widespread implications in understanding T4F biology.

Type 4 filaments (T4F) are a superfamily of nanomachines ubiquitous in Bacteria and Archaea, centred on filamentous polymers of type 4 pilins[1,2]. The best known T4F are type 4 pili (T4P)[3] and type 2 secretion systems (T2SS)[4]. T4F mediate a staggering array of different functions such as adhesion, motility (swimming and twitching), DNA uptake, formation of bacterial communities, and protein secretion[1]. T4F have been an important research area for decades because they are virulence factors in many bacterial pathogens[3].

T4F are usually composed of one major and several minor (low abundance) type 4 pilins[5]. These subunits are synthesised with a characteristic N-terminal (NT) class 3 signal peptide (SP3)[6], which needs to be processed by a dedicated prepilin peptidase before T4F assembly[7,8]. The SP3 consists of a hydrophilic leader peptide, followed by a stretch of 20-25 predominantly hydrophobic residues[5] with often a Glu in position 5. In the characteristic "lollipop" structure of full-length pilins, this hydrophobic stretch represents the "stick" that protrudes from a globular head, which is usually centred on an anti-parallel β-sheet[9]. The stick is the NT half of a universally conserved α1-helix of 50–55 residues (α1N), the C-terminal (CT) half of which (α1C) is part of a structurally variable globular head. The first atomic model of a

T4P in *Neisseria gonorrhoeae*[9], based on fibre diffraction and electron microscopy (EM) results, suggested that the α1 helices form a hydrophobic core by packing helically, in a roughly parallel fashion. This model was then refined to 12 Å resolution by fitting the crystal structure of the pilin into a cryo-EM density map[10]. In the past few years, advances in cryo-EM led to multiple near-atomic resolution T4F structures[11], which revealed a conserved helical architecture, albeit with different symmetry parameters. A striking observation in *N. meningitidis* T4P was that a portion of α1N is "melted" (non-helical), which must occur during polymerisation of the pilin subunits into filaments[12]. This feature was subsequently reported in T4P from enterohemorrhagic *Escherichia coli*[13] (EHEC), *Pseudomonas aeruginosa*[14], *N. gonorrhoeae*[14] and *Thermus thermophilus*[15], and in T2SS from *Klebsiella oxytoca*[16] and *Geobacter sulfurreducens*[17].

All the above structures were determined in diderm bacterial species, which were for more than 20 years the only available models to study T4F. It therefore remains to be determined whether this filament architecture is universal in bacteria. Moreover, none of these structures provided a complete picture of the corresponding T4F since they do not encompass minor pilins, which are key but often poorly

[1]Institut Européen de Chimie et Biologie, Université de Bordeaux-CNRS (UMR 5234), Pessac, France. [2]Laboratoire de Chimie Bactérienne, Institut de Micro-biologie de la Méditerranée, Aix-Marseille Université-CNRS (UMR 7283), Marseille, France. [3]MRC Centre for Molecular Bacteriology and Infection, Imperial College London, London, UK. [4]Plateforme de Microscopie, Institut de Microbiologie de la Méditerranée, Aix-Marseille Université-CNRS, Marseille, France. ✉e-mail: vladimir.pelicic@inserm.fr; remi.fronzes@u-bordeaux.fr

characterised players in T4F biology. Recently, phylogenetically distant monoderm bacteria became a promising new T4F research avenue[18,19], which could help answering the above issues. The main monoderm model is *Streptococcus sanguinis*, a commensal of the human oral cavity frequently causing endocarditis. *S. sanguinis* T4P have been characterised in depth[20–24], revealing that they are heteropolymers of two major pilins (PilE1, PilE2)—which is unusual—and three minor pilins (PilA, PilB, PilC). The structure of the globular heads of all these subunits and their functions have been determined, which is yet to be achieved for most T4F. The two major subunits display a canonical pilin fold, with an uncommonly flexible C-terminus[22]. The three minor pilins are predicted to form a complex that localises at the tip of T4P[24] and promotes adhesion to various host receptors via the modular pilins PilB and PilC[23,24]. PilB and PilC are unusually large pilins with grafted modules conferring adhesive properties[23,24], while PilA is an anchor for PilC at the tip of pilus.

In this paper, using the monoderm *S. sanguinis* as a model, we show that the filament architecture with a melted portion of α1N is universal in bacterial T4F. We produce a complete picture of *S. sanguinis* T4P with all its subunits, and we discuss the wider implications of our findings for the T4F superfamily of nanomachines.

## Results

### T4P on *S. sanguinis* are flexible filaments 7 nm in width

We previously reported that *S. sanguinis* T4P in highly pure pilus preparations exhibit two different morphologies: thick/wavy (12–15 nm) and thin/straight (6–7 nm)[20,22]. To determine the morphology of T4P on the surface of bacterial cells, we observed *S. sanguinis* by transmission EM (TEM) after negative staining. To facilitate observation of T4P, we used a *Δfim* mutant where we deleted the *fim* locus, which is involved in the production of the unrelated sortase-assembled pilus[25]. As can be seen in Fig. 1, the T4P emanating most often from the old poles of *S. sanguinis* cells exhibit a classical T4P morphology[1]. They are up to several μm in length and ~7 nm in diameter. Often, a few filaments emanating from the same cellular locations aggregate laterally at their bases (Fig. 1), but they do not form large bundles like T4P in most diderm species[1]. Therefore, in pilus preparations, the 6–7 nm-wide filaments correspond to native *S. sanguinis* T4P.

To determine the structure of *S. sanguinis* T4P, we analysed purified pili by cryo-EM. Three distinct types of filaments were observed on the micrographs: the two different forms of T4P previously seen by TEM[20,22]—6–7 nm and 12–15 nm in diameter—as well as very thin filaments (3 nm-wide) (Fig. S1a). A preliminary structural analysis of the latter two filaments allowed us to exclude them from further analysis. In brief, for the 12–15 nm-wide filaments, we generated a low-resolution

electron density map after 2D classification, which revealed a cylindrical structure with no distinctive features (Fig. S1b). It remains thus unknown whether these thick T4P result from a dramatic change in quaternary conformation of the filaments (as reported for *N. gonorrhoeae* T4P[26]) or from their limited denaturation upon shearing from the bacterial surface. In contrast, for the 3 nm-wide filaments, we could produce a density map at 6 Å resolution in which the structure of B-DNA could be fitted readily (Fig. S1c), indicating that these thin filaments might correspond to DNA. This was also reported in cryo-EM studies of other T4F[17,27,28]. This extracellular DNA, which is unlikely to result from cell lysis—because no ribosomes were observed in the pilus preparations—is probably actively released by *S. sanguinis*, a well-known property of this species[29].

We therefore focused on the ~7 nm-wide filaments, which look like the T4P seen on the surface of *S. sanguinis*. We segmented these filaments and performed iterative rounds of 2D classification followed by ab initio reconstruction (Fig. S2). After 3D refinement and local refinement on the central portion along the filament axis (150 Å long), we generated a final density map at 3.7 Å resolution (Fig. 2), as estimated by gold-standard Fourier shell correlation (FSC) (Fig. S3a, b). The resolution could not be improved by performing additional 3D classification/variability experiments. The reconstructed filament is a cylinder, 70 Å in diameter, with significant curvature (Fig. 2a). The resolution, which is close to 3.3 Å in the central portion of the filament, decreases to 4.3 Å towards the edges, especially towards both ends (Fig. 2b). This made it impossible to precisely calculate the helical symmetry operators for the filament, which could nevertheless be estimated at 11 Å in rise and 93° in twist angle. The 3D reconstructions were not improved further upon imposition of these helical symmetry operators (Fig. S3c, d).

### *S. sanguinis* T4P are composed of two major subunits arranged stochastically

In contrast to the other characterised bacterial T4F, *S. sanguinis* T4P are heteropolymers composed of two major pilins in comparable amounts, PilE1 and PilE2[20,22], which brings the question whether these subunits alternate regularly in the filaments or are distributed stochastically.

PilE1 and PilE2 show extensive sequence identity (Fig. 3a). The main distinctive feature is an extra 8-aa loop at the end of the αβ region in PilE1 (between residues 92 and 100), which makes this protein slightly longer than PilE2 (Fig. 3a). Within the final density map, individual pilin subunits could be readily identified (Fig. 3b). Strikingly, the regions of difference between PilE1 and PilE2 (Fig. 3c)—a CT "tail" and the 8-aa loop in PilE1—correspond to areas of significantly lower quality in the density map (Fig. 3c). This precluded us from distinguishing PilE1

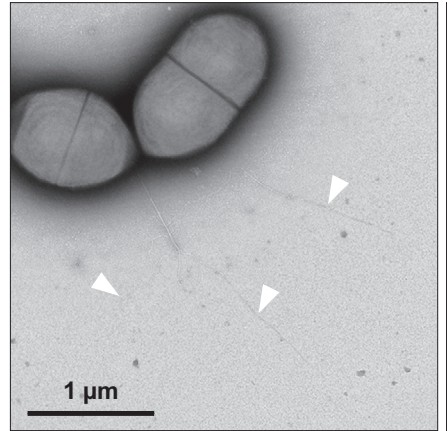
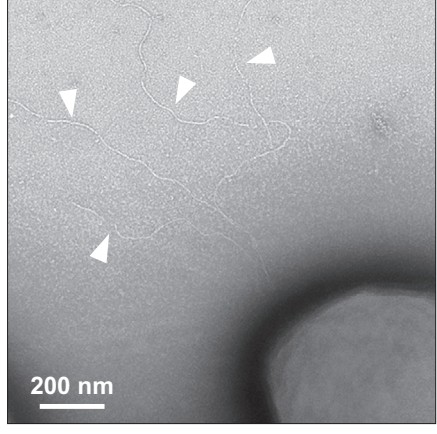

**Fig. 1 | T4P on *S. sanguinis* cells.** Filaments were imaged by TEM after negative staining. Two representative images, at different scales, are shown (experiment was repeated three times). Individual filaments, often emanating from the same cellular locations, are indicated by white arrowheads.

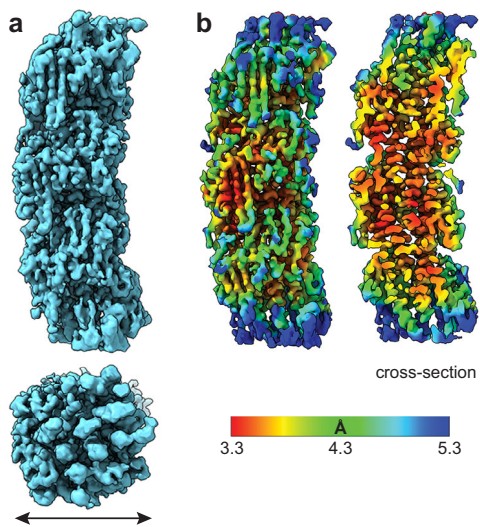

**Fig. 2 | Cryo-EM density map of *S. sanguinis* T4P at 3.7 Å resolution.** The map was obtained without imposing a symmetry. **a** Final density map sharpened with DeepEMhancer[37]. Side (top) and end (bottom) views of the filament are presented, with its diameter indicated. **b** Final density map coloured according to local resolution. A cross-section is shown on the right.

and PilE2 subunits. At the resolution of our map, it should have been possible to model the 8-aa loop, which in contrast could not be modelled at all. The possibility that the lower quality in the density map might be due to an inherent flexibility of the 8-aa loop can be a priori excluded by the previously determined NMR structure of PilE1, which showed no flexibility in this loop within the NMR ensemble[22].

Recently, the first structure of a heteropolymeric T4F has been reported—the archaellum (archaeal flagellum) from *Methanocaldococcus villosus* —in which the two major subunits ArlB1 and ArlB2 alternate regularly[30]. Such filament assembly was proposed to result from the polymerisation of pre-formed ArlB1ArlB2 heterodimers, since the two proteins interact preferentially with one another[30]. We therefore tested the interactions between *S. sanguinis* PilE1 and PilE2 using the bacterial adenylate cyclase two-hybrid (BACTH) system[31], which has proven effective for many T4F proteins[32,33]. For each pilin, we generated two different BACTH plasmids by fusing full-length PilE1 and PilE2 at the NT of T18 and T25 domains of *Bordetella pertussis* adenylate cyclase. We then assessed functional complementation between all possible pairs of T18 and T25 plasmids by co-transformation in an *E. coli cya* mutant and plating on selective indicator plates. As can be seen in Fig. 4a, all plasmid combinations yielded coloured colonies, indicating that PilE1 and PilE2 interact. The efficiency of the functional complementation was quantified by measuring β-galactosidase activities (Fig. 4b). In contrast to what was reported for ArlB1 and ArlB2 in *M. villosus*[30], we found that PilE1 and PilE2 interact equally well with themselves as with one another. This argues against the possibility that *S. sanguinis* T4P could be heteropolymers in which the two major subunits would alternate regularly upon polymerisation of pre-formed PilE1PilE2 heterodimers.

Taken together, our findings are in favour of stochastic assembly of PilE1 and PilE2 within *S. sanguinis* T4P.

### *S. sanguinis* T4P are canonical bacterial T4F

The resolution of the map was better than 3.4 Å in most parts (Fig. 2b), allowing an accurate building of the filament. The resolution was highest in the central portion of the filament, which corresponds to the α1 helices. This allowed us to fit readily the peptide backbone of that portion of the pilin—identical in sequence in PilE1 and PilE2 (Fig. 3a)—within the representative cryo-EM densities (Fig. 5a). Fitting was

unambiguous and large sidechains were readily visible. Critically, in each subunit the α1 helix is interrupted by an unfolded stretch (Fig. 5a), showing that this feature is conserved in all bacterial T4F structures determined so far[11].

As PilE1 and PilE2, which could not be distinguished within the final density map, share high sequence identity, we arbitrarily chose PilE1 to produce an atomic model of *S. sanguinis* T4P. Copies of PilE1 generated by AlphaFold[34] were manually docked into the final density map. The core of the globular part of the pilins (residues 28 to the end) fitted readily in the densities. For the rest of the atomic model, flexible fitting of each subunit and optimisation of the model geometry was performed using ISOLDE[35] in ChimeraX[36], against a sharpened map obtained with DeepEMhancer[37]. The final model was refined in Phenix[38] against a map automatically sharpened using the same program. In our atomic model, the pilin subunits display a canonical lollipop structure in which the α1N portion of a 52 residues-long α1 helix protrudes from a globular head (Fig. 5b). In the globular head, α1C is packed against a three-strand β-sheet, which is flanked at its N- and C-termini by two regions with limited secondary structure elements, a large αβ-loop and a short CT tail, respectively (Fig. 5b). Critically, as in other structures of bacterial T4F[11], a portion of α1N is melted (Fig. 5b), the extent of which varies slightly between $Ile_{16}$ and $Gln_{28}$ in the different chains. The superposition of the structures of the complete pilins that we were able to resolve, shows that the α1N up to the end of the melted region are intrinsically flexible, emanating at different angles from the globular heads (Fig. S4), which in contrast remain almost unchanged, aligning with 1.3 Å root-mean-square deviation (RMSD) globally.

The overall architecture of *S. sanguinis* T4P conforms to the shared structural principles in T4F. In brief, the hydrophobic core of the filament consists of a bundle of α1 helices, leaving the opposite face of the globular heads to form the outer shell (Fig. 5c). The filaments are right-handed with four pilins per turn, displaying approx. 11 Å rise and 93° twist between consecutive subunits within the 1-start helix (Fig. 5c). The interactions between subunits that hold the filament together involve residues in various parts of the pilin (Fig. S5) and can be recapitulated at the levels of the 1-start, 3-start and 4-start helices (Fig. 6). Perhaps the most unusual interaction is within the right-handed 1-start helix, where the CT tail in PilE1 (mostly negatively charged) engages the next pilin αβ-loop (mostly positively charged) through electrostatic interactions (Fig. 6a). This stabilising interaction therefore resembles a "Velcro" closure mechanism. In contrast, the CT tail is highly flexible in the structure of PilE1 monomers previously determined by NMR[22]. At the level of the 1-start helix, there are also hydrophobic contacts between the α1N helices of the $S_{-1}$, S and $S_{+1}$ subunits. $Phe_1$, $Leu_6$ and $Val_9$ of subunit S interact with $Phe_1$ of $S_{+1}$, and $Ile_{12}$ and $Ile_{13}$ of $S_{-1}$ (Fig. 6a).

At the level of the left-handed 3-start, the melted region in subunit S is held in a groove between α1N of subunit $S_{+3}$, and α1C of subunit $S_{-3}$ (Fig. 6b). The $Ile_{21}$ and $Ile_{24}$ residues of subunit S establish hydrophobic contacts contact with $Val_4$, $Ile_7$, $Val_8$ and $Ile_{11}$ of subunit $S_{+3}$, helping to break the helix symmetry around $Pro_{22}$ (Fig. 6b). The hydrophobic interactions between S and $S_{+3}$ is favoured by the other side of the groove, the back of α1C of $S_{-3}$, which is mostly negatively charged (Fig. 6b). $Phe_1$ of subunit $S_{+3}$ is thus held between charged residues $Arg_{41}$ of $S_{-3}$, and $Glu_5$ of $S_{+3}$ (Fig.6b). Critically, $Glu_5$—whose essential role in T4F assembly[5] remains incompletely understood[39]—defines together with $Arg_{41}$ (and $Thr_2$), a charged path at the centre of the pilus core. The electrostatic interactions between these three residues are strengthened by the hydrophobic shell that surrounds them. Finally, in the 4-start helix, the $Ile_{21}$ and $Ile_{24}$ residues in subunit S also establish hydrophobic interactions with $Val_{44}$ and $Ile_{51}$ in $S_{-4}$ (Fig. 6c). In addition, $Gln_{28}$ in subunit S establishes electrostatic interactions with $Thr_{58}$ at the top of the α1 helix in subunit $S_{-4}$ (Fig. 6c).

Taken together, these findings indicate that *S. sanguinis* T4P are canonical bacterial T4F, i.e., the centre of the filament is formed by the

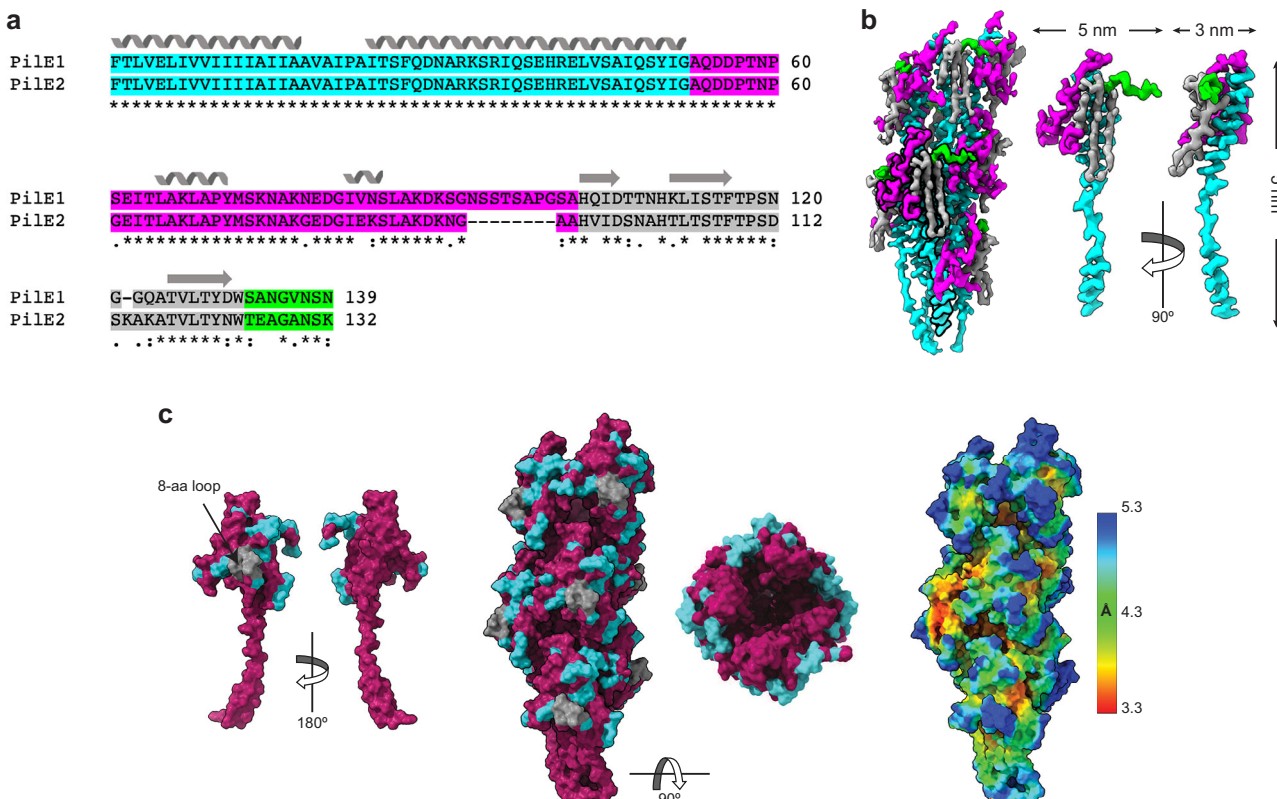

**Fig. 3 | *S. sanguinis* T4P are composed of two major subunits arranged stochastically. a** Sequence alignment between PilE1 and PilE2 using ClustalOmega[58]. (*) conserved aa; (:) aa with strongly similar properties; (.) aa with weakly similar properties. The NT α1-helix, αβ-loop, antiparallel β-sheet, and CT tail are coloured in cyan, magenta, grey and green, respectively. The secondary structures are indicated above the sequences. **b** Pilins can be readily visualised in the density map. The same colour code is used as in **a**. **Left**, final density map where a pilin subunit has been highlighted in bold. **Right**, this individual pilin is shown in orthogonal orientations. The protruding portion of the α1 helix is clearly interrupted by an unfolded stretch. **c** PilE1 and PilE2 cannot be distinguished in the density map. The degree of sequence conservation between the two pilins was mapped with ConSurf[59] on an individual pilin (**Left**) and the filament (**Middle**). The regions that differ between PilE1 and PilE2 are in light blue (with the exception of the 8-aa loop which is in grey) whereas the regions that are conserved are in purple. The regions that differ between the two pilins correspond to areas of poorer local resolution in the 3.7 Å cryo-EM density map of *S. sanguinis* T4P (**Right**).

packing of α1 helices, a portion of which is unfolded. Since these structural features—previously observed in T4F in diderm species—are conserved in a distant monoderm species such as *S. sanguinis*, they are likely to be universal in Bacteria.

## Full structural model of *S. sanguinis* T4P encompassing the minor pilins

We previously characterised the three other subunits of *S. sanguinis* T4P—the minor pilins PilA, PilB and PilC—determining their functions and the structure of their globular heads[23,24]. The modular pilins PilB and PilC were proposed to be located at the pilus tip because their bulky adhesin modules are incompatible with polymerisation in the filament body[23,24]. The non-modular pilin PilA was also predicted to be tip-located since it strongly interacts with (and stabilises) PilC[24]. An AlphaFold[34,40]-computed PilABC model confirmed that the three minor pilins can coexist in one complex. In this helical complex[24]—PilA is added first, interacts with PilC, which interacts with PilB—the grafted adhesin modules in PilB and PilC, cap the pilus[23,24]. We therefore used our filament structure to produce a complete model of *S. sanguinis* T4P by fitting the complex of minor pilins at the tip of the filaments. This was done in two steps.

Since the predicted PilABC architecture implies that PilB connects the complex with the filament, we first positioned PilB in our filament structure. Because PilB and PilE1/PilE2 have canonical SP3 with homologous α1N[22], we produced a PilB model with a portion of α1N melted (Fig. 7a) using SWISS-MODEL[41]. PilB was then positioned into the filament by aligning its α1-helix with the α1-helix of the S5 subunit

(counted from the base) rather than the subunit at the apex, to allow the following subunits to serve as references for aligning the PilAC complex. The alignment of the helices in PilB and PilE1/PilE2 was perfect (Fig. 7b). Next, since the PilAC structural model has been experimentally validated[24], it was docked as such by aligning the α1-helix of PilC—which also exhibits a canonical SP3[22]—with the α1-helix of the $S_6$ subunit (Fig. 7c). Strikingly, this resulted in the α1-helix of PilA aligning with the α1-helix of the $S_7$ subunit and maintaining the helical symmetry of the pilus, which strengthens the validity of this model (Fig. 7c). Finally, by removing $S_5$ and the subsequent major pilin subunits, we produced the final model of a PilABC-capped T4P (Fig. 7d). Critically, although PilA is the first subunit from top, the pilus is capped by the adhesin modules in PilB and PilC, which resemble open wings[24].

By providing a complete picture of a T4F at near atomic resolution, including all its pilin subunits, our model of *S. sanguinis* T4P predicts how minor pilins may cap the pilus tip. In *S. sanguinis*, this architecture is expected to optimise the presentation of the adhesin modules in PilB and PilC for binding to host cells and structures[23,24]. This has direct implications for most T4F, which are similarly capped by complexes of minor pilins, playing a variety of roles extending well beyond adhesion.

## Discussion

T4F—a superfamily of filamentous nanomachines ubiquitous in prokaryotes—have been studied for 40 years, primarily in a handful of closely related diderm species. However, mechanistic aspects of their intricate biology remain poorly understood. which led to the

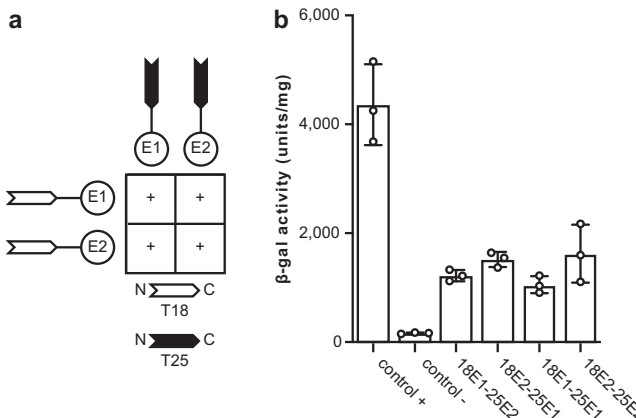

**Fig. 4 | Testing the interactions between PilE1 and PilE2 using BACTH.** The T18 and T25 domains of *B. pertussis* adenylate cyclase were fused to the N-terminus of the full-length proteins. T18 and T25 plasmids pairs were co-transformed in an *E. coli cya* mutant, before plating on selective indicator plates. **a** Combinations of T18 and T25 plasmids producing coloured colonies (+), which indicates functional complementation between T18 and T25, owing to an interaction between the corresponding PilE1 or PilE2 proteins. **b** The efficiency of the functional complementation, which reflects the affinity of the interacting proteins for each other, was quantified by measuring β-galactosidase activities (U/mg). The results are the average ± SD from three independent experiments. Source data are provided as a Source Data file.

development of phylogenetically distant models, recently opening new research avenues. In particular, the monoderm bacterium *S. sanguinis* emerged as a frontline T4F model[19] because all its major and minor pilin subunits have been structurally and functionally characterised[22–24]. In this study, we have used cryo-EM to determine the structure of *S. sanguinis* T4P, which sheds new light on T4F by leading to the findings discussed below.

We have determined the first T4F structure from a monoderm species, which is also the only known heteropolymeric T4F structure in bacteria. This revealed that *S. sanguinis* T4P share the structural characteristics that define T4F, which is not unexpected, but display unique and intriguing features that are likely to play important functional roles. Like in all other T4F, pilin subunits in *S. sanguinis* T4P are arranged in a helical array, in which the extended NT α1-helices form the filament core, while the opposite face of the globular heads form the filament outer shell. Critically, in each pilin subunit, a portion of α1N helix is melted. This structural feature, which was previously observed in T4F from diderm species[12–17] that are phylogenetically distant from *S. sanguinis*, is therefore likely to be universal in Bacteria. Interestingly, in *S. sanguinis* T4P, the α1N is intrisically flexible. Although it remains merely a hypothesis, this property might play a role in the inherent flexibility of the filaments. However, although the bacterial T4F architecture is conserved overall, there are significant structural differences among pilins in the number of β-strands in the central β-sheet and size/structure of the flanking αβ-loop and CT regions (Fig. 8). This leads to differences in pilin shapes and sizes, which results in diverse T4F width and helicity parameters (Fig. 8). Interestingly, while in diderms the pilin globular heads are tightly compacted within the filaments, in *S. sanguinis* T4P they are more loosely connected with gaps between them. It is possible that this property contributes to the intrinsic flexibility of *S. sanguinis* T4P. Such flexibility is expected to have an impact on the T4P-mediated properties—twitching motility[20] and adhesion to host cells[23,24]—by facilitating the movement of bacteria and enhancing interaction with surfaces and/or host cells.

An important consequence of the above structural organisation is that the surface of T4F is predominantly constituted by the two edges that flank the central β-sheet and differ the most between pilins,

namely the αβ-loop and CT regions (Fig. 8). T4F surfaces are often further diversified by post-translational modifications (PTM)[42]. This variability is thought to have important consequences on T4F-mediated functions and is likely used to evade the host immune response[43] or predation by phages[44]. Although *S. sanguinis* T4P share this structural organisation, they exhibit significant differences. The first difference concerns the unstructured 10 aa-long CT tail in *S. sanguinis* major pilins. In monomers, as previously shown by NMR[22], this tail exhibits a variety of conformations and orientations and is therefore highly flexible, while the rest of the protein is not. This was a puzzling observation since in diderms T4F piliation occurs only when this CT region is stabilised by being "stapled" to the last β-strand in the central β-sheet, either by a disulfide bond (hence its well-known "D-region" moniker) or by coordination of a metal[5]. Instead, our structure of *S. sanguinis* T4P reveals that the CT tail of pilin subunits is stabilised only upon polymerisation within a filament, via a completely different mechanism. In brief, the CT tail in one subunit attaches to the αβ-loop of the next through a Velcro mechanism, involving a series of electrostatic interactions. The second difference is that unlike in diderms, there are no PTM on the edges in *S. sanguinis* major pilins[22], but the surface of the pilus is diversified in radically different fashion, i.e., by the stochastic polymerisation of the two major pilins in the pilus. This would ensure that the pilus surface is mosaic, which should promote better immune evasion than a regular arrangement of the two major pilins. In addition, as proposed for the heteropolymeric T4F in *M. villosus*[30], a heteropolymeric pilus could confer other advantages by modulating filament stability and/or T4P-mediated functions. Accordingly, in *S. sanguinis*, single *ΔpilE1* and *ΔpilE2* mutants are piliated (piliation is abolished only in the double *ΔpilE1ΔpilE2* mutant), but they produce less filaments than the parental strain[20]. Moreover, although they exhibit twitching motility these single mutants move either slower (*ΔpilE1*) or faster (*ΔpilE2*) than the wild-type strain[22].

The fact that all known T4F are composed of major and minor pilins[2] makes our complete structural model of *S. sanguinis* T4P—encompassing all the minor pilins that are key players in T4F biology—of general significance. We could achieve this because, unlike in most other systems, all the minor pilins of *S. sanguinis* T4P (PilA, PilB, and PilC) have been previously characterised structurally and functionally[22–24]. Our atomic model has implications for most (if not all) T4F, but there is an important caveat: it is merely a prediction and therefore requires further experimental validation. For example, the adhesin modules in PilB and PilC are almost certainly highly flexible because of the unstructured loops that connect them to their pilin modules[24]. Therefore, the PilB and PilC wings are expected to be "flapping", which should maximise bacterial adhesion. Second, although it is data-driven, the proposed architecture of the pilus tip and the PilABC complex remains an educated guess. Indeed, because its SP3 is similar to that of PilE1, we opted to model PilB with a melted α1N portion. In contrast, because the interaction interface between the globular domains of PilA and PilC has been experimentally validated by NMR[24], we decided to dock the PilAC AlphaFold model as such. However, since the SP3 of PilC is similar to PilE1 it cannot be excluded that the α1N of PilC too is melted upon filament polymerisation, which is unlikely for PilA that has a highly peculiar SP3[22]. Determining an atomic resolution structure of a T4F with its tip-located minor pilins would shed light on these issues, but this would require methodological and technical advances beyond the state of the art. These advances include pilus preparation methods that would preserve the integrity of the filaments from base to tip, or new tomography tools allowing the structural characterisation of filaments emanating from cells at near-atomic resolution.

The complete model of *S. sanguinis* T4P, including three minor pilins at the tip, has several important implications. The filament starts with PilA, the only pilin that lacks a Glu$_5$. This is consistent with the structural characterisation of a complex of four minor pilins

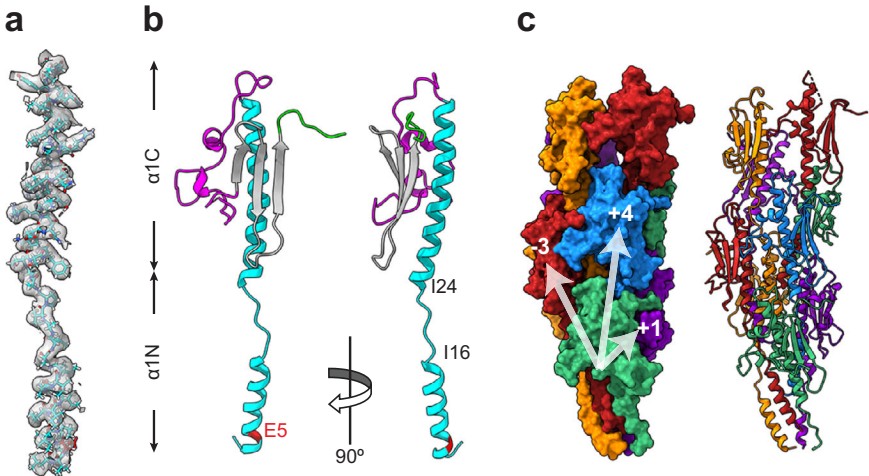

**Fig. 5 | Atomic model of *S. sanguinis* T4P.** This model was built using an AlphaFold-predicted PilE1 stucture. **a** Fitting of the PilE1 NT portion (stick representation) within the representative cryo-EM densities in the final density map. This portion is identical in PilE2. **b** Ribbon representation of a PilE1 subunit in the filament. The positions of Glu5 and the melted region between Ile16 and Ile24 are

indicated. We used the same colour code as in Fig. 3. **c** *S. sanguinis* T4P structural model. Surface (left) and ribbon (right) representations with the individual pilins outlined in different colours. Connectivity is shown in the right-handed 1-start (+1), right-handed 4-start (+4), and left-handed 3-start (-3) helices.

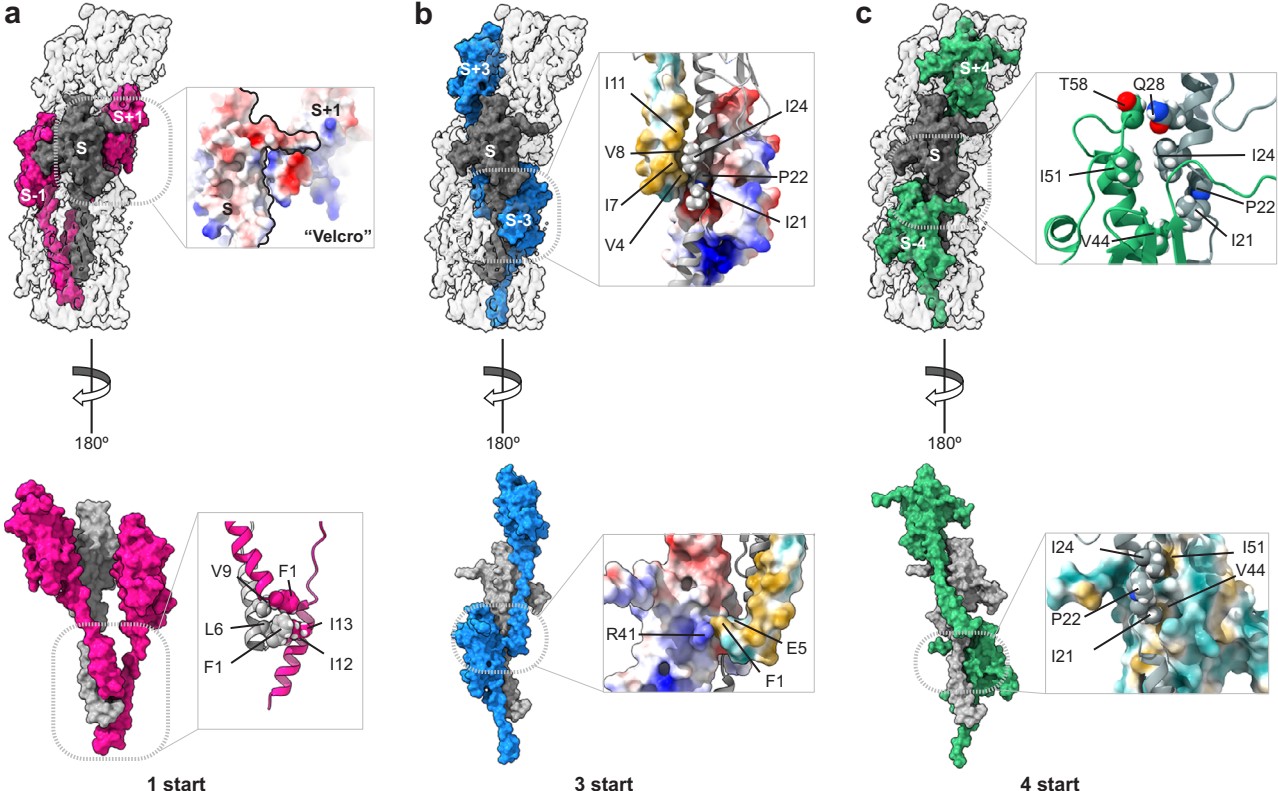

**Fig. 6 | Interactions between the subunits in *S. sanguinis* T4P. a** Interactions between the S (grey) and $S_{+1}/S_{-1}$ subunits (pink) in the 1-start helix (180° views). **Top**, enlarged view shows a surface representation of the electrostatic potential of the S and $S_{+1}$ subunits that interact via a Velcro mechanism. **Bottom**, enlarged view of the interactions between the NT parts of the three pilins (ribbon representation), with the residues involved highlighted. **b** Interactions between the S (grey) and $S_{+3}/S_{-3}$ subunits (blue) in the 3-start helix (180° views). **Top**, enlarged view of the interactions between the S (ribbon representation), $S_{+3}$ (hydrophobicity surface representation) and $S_{-3}$ (electrostatic potential surface representation) subunits, with the

residues involved highlighted. **Bottom**, enlarged view of the interactions between the $S_{+3}$ (hydrophobicity surface representation) and $S_{-3}$ (electrostatic potential surface representation) subunits, with the residues involved highlighted. **c** Interactions between the S (grey) and $S_{+4}/S_{-4}$ subunits (green) in the 4-start helix (180° views). **Top**, enlarged view of the interactions between the S and $S_{-4}$ subunits (both in ribbon representation), with the residues involved highlighted. **Bottom**, enlarged view of the interactions between the S (ribbon representation) and $S_{-4}$ (hydrophobicity surface representation) subunits is represented, and the interacting residues are indicated.

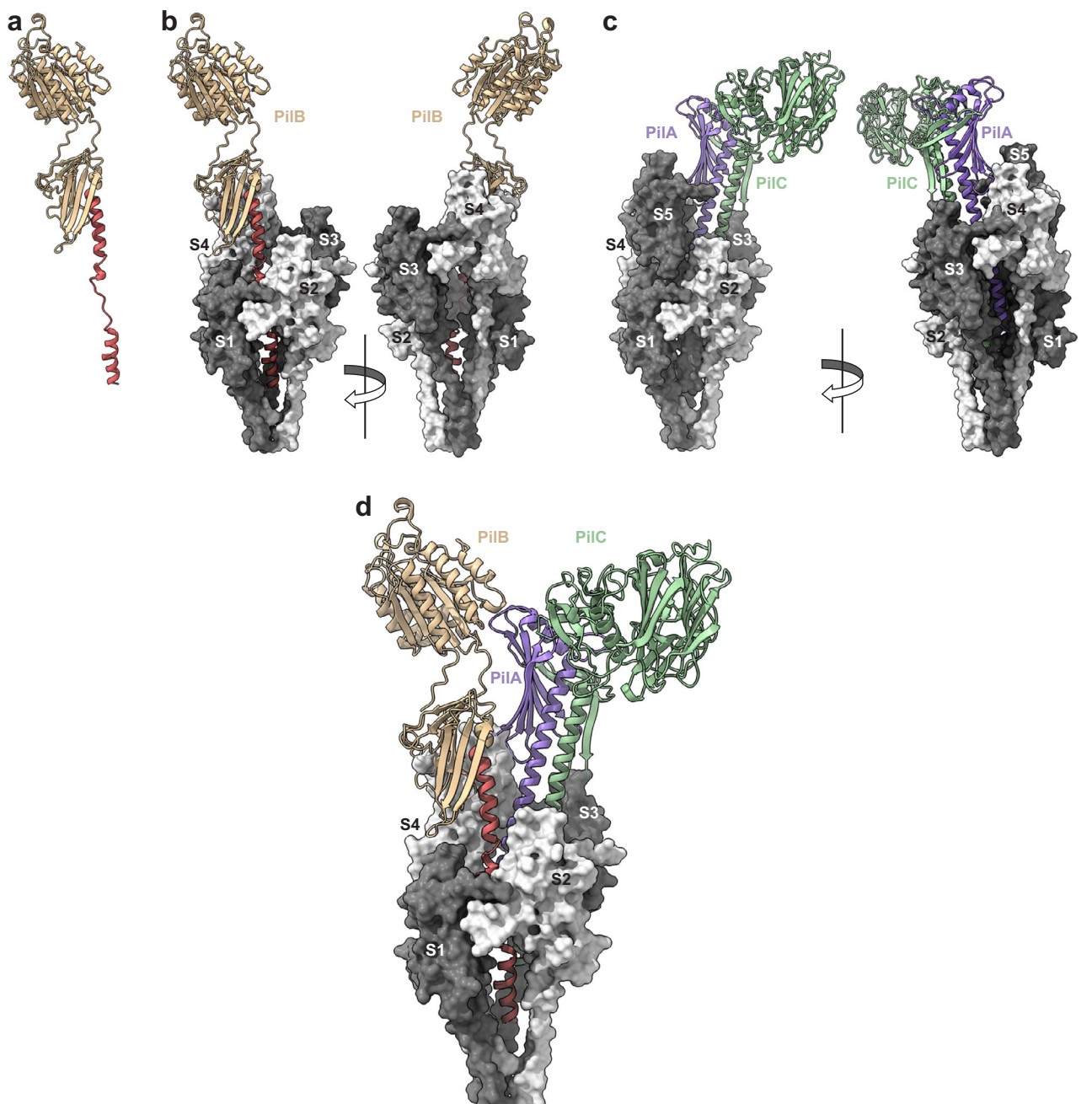

**Fig. 7 | Integrated atomic model of the heteropolymeric T4P in *S. sanguinis* with its tip-located complex of minor pilins. a** PilB model with a portion of α1N melted. **b** PilB superposition with PilE1 subunit S5 in the filament structure (180° views). **c** Superposition of PilAC with PilE1 subunits S6 and S7 in the filament structure (180° views). **d** Integrated model of the T4P pilus in *S. sanguinis*.

widespread in T4F[45–47], generically known as HIJK[48], which is capped by the K subunit that also lacks Glu5. Since the negatively charged Glu5 in subunit S usually establishes an important salt bridge with the positively charged N-terminus of the S+1 subunit, it is not unexpected that this residue is not needed (and therefore absent) in the subunit at the apex of the pilus, because there is no subunit above it. Since PilA specifically interacts and stabilises PilC[24], this is the next pilin in the filament. PilC has an unusually large modular pilin with a lectin domain, which bind glycans prevalent in the human glycome[24]. The AlphaFold model of PilAC, which was consistent with the interaction interface characterised experimentally[24], fitted very well in the filament structure. PilA is thus an anchor for PilC, facilitating the presentation of this adhesin at the tip of filaments. The structural homology between PilA and the I subunit of the HIJK

complex[24] suggests that this latter complex might play a similar role—facilitating the presentation at the tip of effectors such as PilC/PilY1 in T4P from diderms[49], or secreted effectors in T2SS[50]. The third pilin in the filament is PilB, another large modular pilin and bona fide adhesin with a vWA module that binds protein ligands such as fibronectin and fibrinogen[23]. The AlphaFold model of PilB, in which we melted a portion of α1N, fitted very well on its own in the filament structure. This led to significant remodelling of the interface between PilB and PilC proposed by AlphaFold in the PilABC complex[24], which interestingly could not be fitted as such. This is reminiscent of the melting in α1N of pilins occurring during filament assembly, which currently cannot be predicted by AlphaFold but was demonstrated in multiple T4F[12–17] by cryo-EM, including in *S. sanguinis* in this study.

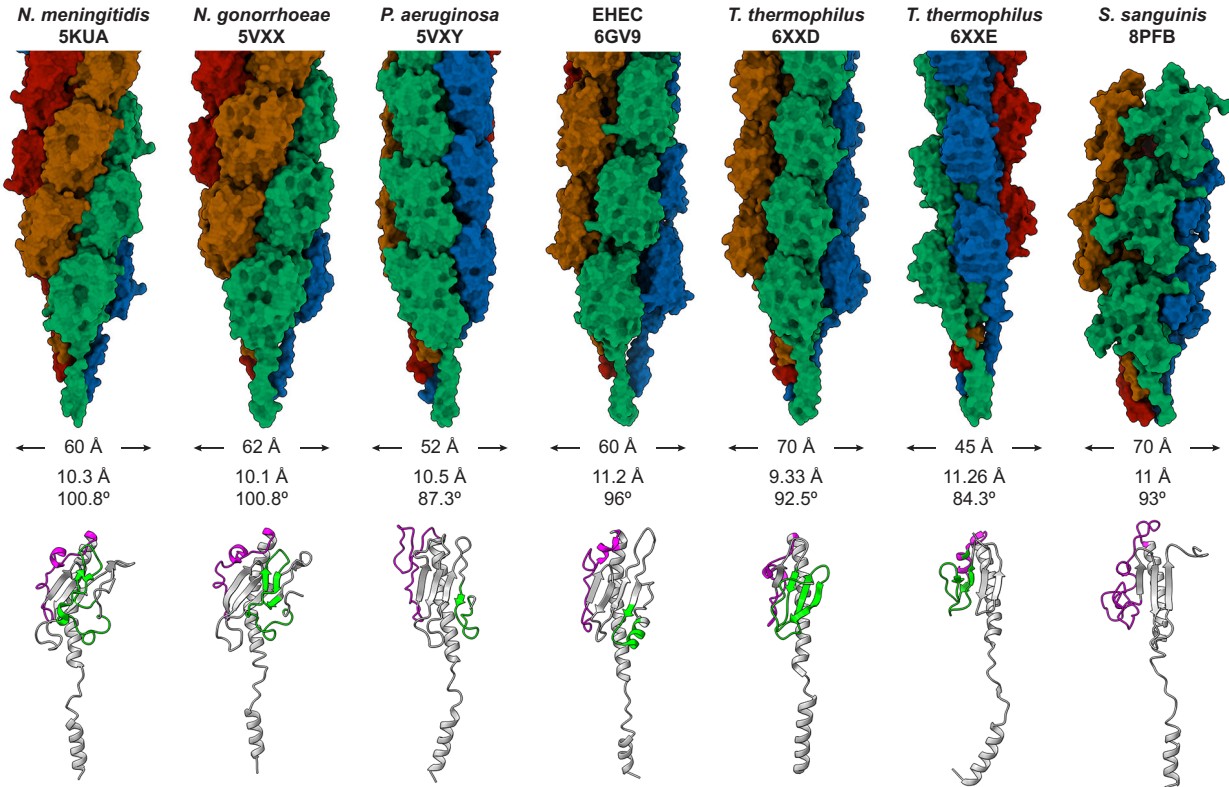

**Fig. 8 | Comparison of known bacterial T4P structures.** From Left to right, *N. meningitidis*[1], *N. gonorrhoeae*[2], *P. aeruginosa*[2], EHEC[3], *T. thermophilus*[4] (PilA4 and PilA5 T4P) and *S. sanguinis* (present work). **Top**, pilus structure with the pilins coloured along the 4-start helix. The PDB identifiers, diameter and helical symmetry operators are indicated for each filament. **Bottom**, structure of pilin subunits. The pilins have been extracted from the PDB structure of the corresponding filament. For each pilin, the αß-loop is coloured in magenta and the D-region (if present) in green.

In conclusion, by solving the first structure of a T4F in a monoderm bacterium and producing a complete picture including the minor pilins (key but often poorly characterised players in T4F biology), this study has general implications for T4F. Moreover, it cements *S. sanguinis* as a major T4F model, paving the way for further investigations that will improve our understanding of these fascinating filaments.

## Methods

### Strains and growth conditions

*E. coli* strains were grown in liquid or solid lysogeny broth (LB) medium (Difco), with spectinomycin (100 µg/ml), kanamycin (50 µg/ml) and/or ampicillin (100 µg/ml), when required. All antibiotics were from Sigma. Strain DH5α was used for cloning, while strain BTH101 (Euromedex)—a non-reverting *cya* mutant—was used in BACTH assays[31]. We amplified the full-length genes *pilE1* (5'- cgcggatccCATGTTAAACAAATTACAAA AATTCCG−3' and 5'-cgcggtaccGCGTTTGAGTTTACACCATTAGCA−3') and *pilE2* (5'- cgcggatccCATGTTAAACAAATTGCAAAAATTCCG−3' and 5'- cgcggtaccGCTTTTGAATTAGCACCAGCTTC−3') from genomic DNA of *S. sanguinis* 2908[20] using high-fidelity Pfu DNA polymerase (Agilent). PCR products were cloned directly into pCR8/GW/TOPO (Invitrogen). Inserts were verified by sequencing, cut out from the TOPO derivatives by *Bam*HI and *Kpn*I digestion, gel-extracted, and subcloned into the BACTH vectors pUT18C and pKT25. Cloning was carried out using standard molecular biology techniques[51].

*S. sanguinis* was grown as described[20] on plates containing Todd Hewitt (TH) broth (Difco) and 1% agar (Difco), or in liquid culture in THTH, i.e., TH broth containing 0.05 % tween 80 (Merck) to limit bacterial clumping, and 100 mM HEPES (Euromedex) to prevent acidification of the medium. When required, 500 µg/ml kanamycin was used for selection. Plates were incubated at 37 °C in anaerobic jars (Oxoid) under anaerobic conditions generated using Anaerogen sachets (Oxoid), while liquid cultures were grown statically under aerobic conditions. To construct the *S. sanguinis Δfim* mutant, we deleted the locus involved in the production of sortase-assembled pili[25]. We replaced the genes from SSV_1499 (*srtC*) to SSV_1503 by a promoterless *aphA-3* cassette, which confers resistance to kanamycin. To do this, we fused by splicing PCR the regions upstream (5'-GCCAAGCACCTGACTAGTAG-3' and 5'-ggtgatattctcatttttagccatTATAATCTCCTAATTTTATCTTCACTC-3') and downstream the *fim* locus (5'-ttttactggatgaattgttttagGGAAAAGAAAAGAG CCGAGC-3' and 5'-ATTCCACCGCGTCATCAATG-3') to *aphA-3* (5'-A TGGCTAAAATGAGAATATCACC-3' and 5'-CTAAAACAATTCATCCAGT AAAA-3'). We directly transformed the PCR product into strain 2908 and selected allelic exchange mutants on kanamycin plates. Allelic exchange was confirmed by PCR.

### T4P visualisation

Surface-associated T4P in *S. sanguinis Δfim* were visualised by TEM after negative staining as follows. Bacteria were grown in THTH until OD$_{600}$ reached 0.8, adsorbed for 3 min to glow-discharged carbon-coated grids (EMS), and fixed 5 min in 2% glutaraldehyde. The grids were cleaned by floating them sequentially 10 times on drops of pilus buffer (20 mM Tris, pH 7.5, 50 mM NaCl), and then stained for 2 min with 2% aqueous uranyl acetate. Stain solution was gently drained off the grids, which were air-dried before visualisation using a Tecnai 200 KV electron microscope (Thermo Fisher Scientific). Digital image acquisition was made with a 16 megapixel, CMOS, Oneview numeric camera (Gatan).

### T4P purification

T4P were purified from *S. sanguinis* as described elsewhere[20] with minor modifications. Liquid cultures grown O/N in THTH were used to re-inoculate pre-warmed THTH and grown statically until the OD$_{600}$ reached 1. Bacteria were pelleted by centrifugation for 10 min at

4,149 g at 4 °C. Pellets were re-suspended in ice-cold pilus buffer by vigorous pipetting up and down, which was enough to shear T4P. Bacteria were then pelleted as above, and supernatant containing the pili was transferred to a new tube. This centrifugation step was repeated, before the supernatant was passed through a 0.22 μm pore size syringe filter (Millipore). Pili were then pelleted by ultracentrifugation, resuspended in pilus buffer, and tested by SDS-PAGE/Coomassie, essentially as described[20].

## Cryo-EM sample preparation and data acquisition

R2/2 Cu 200 mesh grids (Quantifoil) were glow-discharged for 40 s at 2.7 mA. Then, 4 μl of freshly purified pili were applied and the excess of sample was immediately blotted away (3.5 s blot time, 4 °C chamber temperature, 100% humidity) in a Vitrobot Mark IV (Thermo Fischer Scientific), before being plunge-frozen in liquid ethane. Cryo images of the purified pili were recorded with a Talos Arctica microscope (Thermo Fischer Scientific) operated at 200 kV and equipped with a K2 summit direct electron detector (Gatan). Dose fractioned data were collected in a defocus range of −0.4 to −1.4 μm at 45,000 X magnification, corresponding to a pixel size of 0.93 Å, using SerialEM[52]. The total dose of electrons was 50.74/Å², with 1.34 electrons/Å² per frame. The cryo-EM data collection and refinement statistics are detailed in Table S1.

## Image processing

All data processing was carried out in Cryosparc[53]. The cryo-EM data processing workflow is schematised in Fig. S2. In brief, movies were aligned for beam-induced motion using the function "Patch motion correction", while CTF (Contrast Transfer Function) parameters were assessed using "Patch CTF Estimation". Non-overlapping segments of the S. sanguinis T4P were manually picked, and particles were extracted using a box size of 320 pixels. These particles were 2D classified, the best 2D classes were selected and used as references to automatically pick the filament in all the micrographs. This was done using the program "Filament Tracer"[54] by indicating a filament diameter of 70 Å, a separate distance between segments of 42 Å and a minimum filament length of 60 Å. After extraction, several rounds of 2D classification were performed, and 393,556 particles corresponding to well-resolved classes were selected for further processing. Four ab initio models were generated and the best one was used as 3D reference for homogeneous refinement. A mask was created using the function "Volume Tool" for the central part of the filament, which was used for several rounds of local refinement and global CTF refinement. This processing led to a final map at 3.67 Å of resolution. The Cryosparc symmetry search tool was used to determine the average helical symmetry operators during helical refinement. The initial symmetry operators−93° of twist and 11 Å of rise−were refined to 93.18° and 11.34 Å in the final map. When the processing was redone after imposing these helical symmetry operators, a map at 6.44 Å of resolution was obtained.

For the thick filaments, a similar processing workflow was used. After extraction and several rounds of 2D classification were performed, 109,340 particles were selected for further processing. Several ab initio models were generated and the best one was used for homogeneous refinement. This processing led to a final map at 7.9 Å of resolution.

## Building and refinement of the T4P atomic model

Model of the major pilin PilE1, generated using AlphaFold[34] (pLDDT 78.82), was docked into the refined cryo-EM map using the program "Fit in map" from the software ChimeraX[36]. The map was sharpened in PHENIX[31] (phenix.autosharpen)[55], and the final model was refined by several rounds of manual refinement in ISOLDE[35] and real-space refinement using phenix.real_space_refine[38]. Details about the cryo-EM refinement statistics and FSC Map versus Model plot can be found in Table S1 and Fig. S3b, respectively. The final model was validated using MolProbity[56] and phenix.validation_cryoem[57] implemented in the PHENIX software.

## Testing the interaction between PilE1 and PilE2 by BACTH

BACTH assays as described elsewhere[32] with minor modifications. In brief, BTH101 cells, co-transformed with pairs of recombinant pUT18C and pKT25 plasmids, were plated on selective MacConkey plates supplemented with 0.5 mM IPTG and 1% maltose (Sigma). Plates were incubated at 30 °C and the colour of the colonies was scored after 40 h. Each assay was repeated three times with a positive and a negative control included. The efficiency of the functional complementation between T18 and T25 was quantified by measuring ß-galactosidase activity in liquid culture as previously described[32]. Single colonies were picked from the above MacConkey plates after 48 h of growth, inoculated in 5 ml LB supplemented with 0.5 mM IPTG and antibiotics, and the bacteria were grown O/N at 30 °C. The next day, the cultures were diluted in M63 broth and the $OD_{600}$ was recorded. Cells were permeabilised with chloroform and SDS[32] during 40 min at 30 °C with shaking at 250 rpm. Ten μl of the permeabilised cells were then diluted into 990 μl PM2 buffer containing 100 mM ß-mercaptoethanol, and incubated at 28 °C for 5 min[32]. The ß-galactosidase reaction was started at 28 °C with O-nitrophenol-ß-galactoside diluted in PM2 buffer and stopped with 500 μl 1 M $Na_2CO_3$ after 20 min for positive samples, or 60 min for negative samples. The $OD_{420}$ was recorded, and the enzymatic activity A (units/ml) was quantified as $A = 200 \times (OD_{420}/\text{min}$ of incubation$) \times$ dilution factor. The results were expressed as enzymatic activity per milligram of bacterial dry weight (U/mg), so that 1 unit corresponds to 1 nmol of ONPG hydrolysed per minute at 28 °C[32]. The assay was performed on three independent cultures for each plasmid combination.

## Bioinformatics and modelling

PilB's NT α-helix (residues 1–58) is similar in sequence to the corresponding portion in PilE1 (25% identity). Therefore, to model this part of PilB that is the main assembly interface in the pilus, we performed homology modelling with the NT portion (residues 1–58) of PilE1 using SWISS-MODEL[41]. We then superimposed the NT α-helix in PilB model with the NT α-helix in the $S_5$ PilE1 subunit, close to the centre of our filament structure. Similarly, we superimposed the NT α-helix of PilC in the AlphaFold PilAC model[24] (ipTM+pTM 0.75) with the NT α-helix in the next PilE1 subunit $S_6$, which led to superposition of PilA's NT α-helix with the NT α-helix of $S_7$ PilE1 subunit. The final model was obtained by removing $S_5$ and the subsequent major pilin subunits. The quality of the computed model was estimated using the Structure Assessment tool in SWISS-MODEL, which returned a Global QMEANDisCo score of $0.61 \pm 0.05$.

## Reporting summary

Further information on research design is available in the Nature Portfolio Reporting Summary linked to this article.

## Data availability

The atomic model for the S. sanguinis T4P was deposited at the Protein Data Bank with accession code 8PFB, and the corresponding map was deposited at the Electron Microscopy Data Bank with code EMD-17645. All the data generated during this study are included in this paper and/or its Supplementary Information file. Source data are provided with this paper.

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

## Acknowledgements

This work was funded by the Agence Nationale de la Recherche (ANR-21-CE11-0008-01 to V.P. and R.F.) and the Medical Research Council (MR/P022197/1 to V.P.). We thank Sophie Helaine (Harvard Medical School) and Romé Voulhoux (Laboratoire de Chimie Bactérienne, Marseille) for critical reading of the manuscript.

## Author contributions

V.P. and R.F. were responsible for conception/supervision of the work and writing of the manuscript. R.A., L.P., M.S., H.L.G. and A.K. performed the experimental studies.

## Competing interests

The authors declare no competing interests.
