## [Peer Review File · Nature Communications]

Reviewers' Comments:

Reviewer #1:

Remarks to the Author:

This is an excellent paper that is clearly deserving of publication after only minor revisions. The paper expands our knowledge of type IV pili (T4P) structure, and argues convincingly that properties of these pili (such as a partial melting of the N-terminal helix) are likely to be universal across all bacteria. It might help to mention somewhere that, in contrast, the archaeal proteins containing homologs of the bacterial T4P N-terminal domain, archaeal flagellins and T4P, do not undergo such melting. Here are the minor points that need to be addressed:

Line 94) "predominantly polar". How can they possibly tell from low resolution negative stain images? Most helical polymers (such as F-actin, microtubules, etc.) are polar, meaning that the two ends are different and all subunits have the same orientation in the filament. In contrast, there are some bi-polar polymers (such as muscle thick filaments) where the two ends are identical. But for many of these structures polarity is only evident at high resolution. For F-actin, polarity can be visualized at lower resolution using decoration by myosin S1, generating very polar arrowheads.

Line 94-5) "6.78 nm in diameter". How can the diameter be determined to a precision of a tenth of an Å from low resolution negative stain images? This is meaningless, and it is better to say ~ 7 nm in diameter for these dried and presumably partially flattened filaments.

Lines 104-106) No details have been provided about the image processing of the thick filaments and the reconstruction in Fig. S1B. The paper states that 393,556 particles were used, but this is presumably for the 7 nm filaments. How many were used for the thick filaments?

Line 106-7) It is suggested that the thick filaments may result from a dramatic change in quaternary structure "similarly to what has been reported for *N. gonorrhoeae* T4P" [ref. 26]. But in ref. 26, it is shown that mechanical stretching of the T4P results in much thinner filaments, the exact opposite of what is being described here!

Lines 110-113) The observation of extracellular DNA is very nice, but the authors may be unaware of the extensive discussion of this in a related area, T4P from *Geobacter sulfurreducens*. Many papers, primarily from Lovley and colleagues, have argued that "3 nm" extracellular filaments seen by AFM are T4P formed from only the PilA-N chain, and lacking a globular C-terminal domain. But other labs using cryo-EM have shown that these 3 nm filaments are DNA which is abundant in many preparations (Gu et al., 2021; Wang et al., 2022). This issue is discussed in a recent review (Wang et al., 2023).

Lines 125-126) There is no explanation of exactly what was done for the imposition of helical symmetry (Fig. S2). Was this done with the 393,556 particles? Were the helical parameters allowed to evolve?

Lines 160-164) It is argued that PilE1 and PilE2 are distributed randomly in the filament. It is noted that in an archaeal flagellar filament, two isoforms of the same protein are arranged into hexamers, and not randomly. A very simple observation, the power spectrum from the filament segments, shows that this is the case in the archaeal flagellar filaments, where it is easily seen that the asymmetric unit is a hexamer with six times the axial rise expected for an asymmetric unit containing only one protein subunit. The authors should provide a power spectrum here showing that there is indeed no higher order symmetry arising from a regular arrangement of PilE1 and PilE2.

Line 202) "velcro" needs to be capitalized, as it is a brand name.

Lines 229-230) "We previously determined the structure and function of the three other subunits of *S. sanguinis* T4P, the minor pilins PilA, PilB and PilC". But these structures were not of the whole pilins, only the globular domains.

Line 256) "at atomic resolution". This structure is not at atomic resolution ($\sim 1.2 \text{ \AA}$). Rather, it is at "near-atomic resolution".

Line 257) "our model...is invaluable". This is a level of hyperbole that does not belong in the paper. Further, the model for how the minor pilins cap the filament is only a model, and not an experimental structure.

Line 271) "notable findings" is again hyperbole that could be reduced to "findings".

Line 331) Saying that they do not have an experimental structure and their model "is thus perfectible" implies that a true experimental structure is perfect. It would be much better to simply say that their model needs to be proved or improved experimentally.

Line 358) "AlphaFold model of PilAC, which confirmed the interaction interface characterised experimentally" is not the best phrasing. I do not think that an AlphaFold model can confirm anything. For example, just several sentences later it is said that "This led to significant remodelling of the interface between PilB and PilC proposed by AlphaFold in the PilABC complex." Rather, one might say that the AlphaFold model was consistent with an interaction interface characterized experimentally.

Lines 370-1) "melting in $\alpha 1N$ of pilins occurring during filament assembly, which similarly could not be predicted". AlphaFold predictions are not mystical nor are they based upon physical principles. Rather, they are simply machine learning from a large training set (the PDB). As more structures of T4P with partially melted helices, such as this one, are deposited in the PDB, such melting will be "predicted" in the future.

Lines 373-4) "solving the structure of a T4F in a bacterial species radically different from previously characterized ones" is probably a poor choice of words. I would phrase it simply as a monoderm, whereas all previous T4F structures have been from Gram-negative bacteria.

Line 458) "a separate distance between segments of 0.7 and a minimum filament length to consider of 1." This may have meaning to people who use cryoSPARC but what is a distance of 0.7 and a length of 1?

References

Gu, Y., Srikanth, V., Salazar-Morales, A.I., Jain, R., O'Brien, J.P., Yi, S.M., Soni, R.K., Samatey, F.A., Yalcin, S.E., and Malvankar, N.S. (2021). Structure of Geobacter pili reveals secretory rather than nanowire behaviour. *Nature* 597, 430-434.

Wang, F., Craig, L., Liu, X., Rensing, C., and Egelman, E.H. (2023). Microbial nanowires: type IV pili or cytochrome filaments? *Trends in Microbiology* 31, 384-392.

Wang, F., Mustafa, K., Suci, V., Joshi, K., Chan, C.H., Choi, S., Su, Z., Si, D., Hochbaum, A.I., Egelman, E.H., and Bond, D.R. (2022). Cryo-EM structure of an extracellular Geobacter OmcE cytochrome filament reveals tetrahaem packing. *Nat Microbiol* 7, 1291-1300.

Reviewer #2:

Remarks to the Author:

Type IV pili (T4P) are ubiquitous and highly versatile surface-exposed filaments. So far, only bacterial T4P structures of diderm bacteria have been solved. Moreover, the bacterial structures solved so far are composed of one major pilin subunit or a major pilin subunit composed of two polypeptide chains. These structures have revealed the same overall structural blueprint with a central largely alpha-helical part forming the central core of the pilus, and the pilus surface being formed by more variable globular domains. Here, the authors solve the high-resolution T4P structure from the monoderm *Streptococcus sanguinis* that is heteropolymeric and composed of two different major pilins (Pile1 and Pile2). These two major pilins deviate significantly from other major pilins of solved T4P structures.

The major experimental findings include (A) the high-resolution structure of the *S. sanguinis* heteropolymeric T4P structure, (B) showing that the two major pilins are inserted in random order in the pilus, (C) that the *S. sanguinis* T4P follows the same overall structural blueprint as observed in diderm T4P. Moreover, the authors include an AlphaFold-based structural model of a complex composed of the minor pilins PilA/PilB/PilC. This complex is thought to be at the pilus tip and involved in adhesion. Finally, the authors use their structural data of the *S. sanguinis* T4P and the AlphaFold model of the PilA/B/C tip complex to generate a very nice model of the complete pilus structure.

Altogether, this very nice manuscript with high-quality data and makes important contributions to our understanding of T4P in bacteria. The solved structures are beautiful. Overall, the logic of the presentation is easy to follow. Nonetheless, I have a few comments. These comments are mostly about clarifications and the presentation of data. Only #6 describes an additional experiment to be included.

1. The authors refer to type 4 filaments as nanomachines. To this, I have more of a semantic question: It would seem that T4P per se are not nanomachines but part of a nanomachine that includes all the other proteins of the T4P machine to function. So maybe consider rewriting that T4F are the filamentous parts of a large nanomachine.
2. Line 45: Maybe rewrite to “..structure of mature pilins..”.
3. Line 48: Maybe rewrite to “.. packed against the globular..”.
4. Line 96: Please include arrow(s) in Fig. 1 to indicate the T4P that “aggregate laterally”.
5. Fig. 3B: It is difficult to distinguish the two different subunits. Maybe to show the two different subunits, color them differently (e.g., with shades of cyan, purple, grey and green). Also, please include both pilins in Fig. 3B. Fig. S3A-C very nicely helps to show the differences between the two pilins in the solved structure. Because a major point of the manuscript is that the *S. sanguinis* T4P is composed of two major pilins, I suggest moving Fig. S3A-C to a figure in the main text.
6. Line 155-164: Please include the control experiments in which the four PilE1 and PilE2 constructs are tested separately in the bacterial two-hybrid analyses. These controls are essential to ensure that none of the four constructs self-activate. Concerning the comment that the beta-gal level reflects the strength of binary interactions, this is not entirely correct. This level also depends on the accumulation level of the different fusion proteins. So, if statements are made about the “strength of interactions”, then western blots should be included to demonstrate the level of accumulation of the four proteins.
7. Line 175-191: I have several questions/comments about this para and the relevant figures:
 - (i) In Fig. 3 and S3A-C the solved structures of PilE1 and PilE2 obtained from the pili are shown. Why then are AlphaFold models of PilE1 and PilE2 used to generate the atomic model? If there is a good explanation, please include it.
 - (ii) in the legend to Fig. 5, please include whether the structures shown are solved structures or AlphaFold models. Also, the globular domains are described as having a three-stranded beta-sheet flanked at the N- and C termini with regions “with no distinctive secondary structure”. However, in Fig. 3 and S3, these regions are described as alpha-helical. Please clarify.
 - (iii) Please include a more detailed explanation of what is shown in Fig. S4.
8. For all AlphaFold models, please include the standard quality checks, including pAE and pLDDT diagrams of five models. Please also describe which model was picked for further analysis.

Reviewer #3:

Remarks to the Author:

In this paper, Anger et al describe the structure of a heteropolymeric T4P from the monoderm

bacterium *S. sanguinis*. The structure of the pilus follows the typical T4P architecture, supporting the notion that the canonical T4P architecture is conserved in bacteria. The paper is interesting, as it suggests that the pilus is composed of two pilins, which are stochastically mixed throughout the filament. The authors also suggest a model of the pilus tip formed by minor pilins.

I have the following comments that should be addressed.

Major comments:

1. The authors base their hypothesis that the filament is a random mosaic purely on the fact that an 8 aa loop between residues 92 and 100 is not as well resolved as the rest of the protein. While the hypothesis is intriguing, further evidence should be provided to substantiate it. NMR studies suggesting low flexibility of this area are not sufficiently convicting. Without further experimental support, the interpretation of the map and all derived discussion should be toned down.
2. There is a lot of emphasis on the 8 aa loop between residues 92 and 100. Could the authors provide a figure panel (at least in the supplementary figures) to show this lower resolved area in comparison with the rest of the high-resolution map?
3. The model of the tip is very interesting. However, it is mostly based on an AlphaFold prediction and would thus require experimental verification before substantial functional conclusions are drawn. I appreciate that this would be technically challenging, so therefore, the interpretation of the model needs to be toned down.
4. Furthermore, the AlphaFold model needs to be corroborated by confidence plots.
5. The model of the tip suggests that the hydrophobic alpha helices of PilC and PilA are exposed to the surrounding medium (Fig. 7), which does not seem very convincing. The authors should critically assess this or provide an explanation.
6. It is suggested that the "wings" of the tip flap about and that this enhances the adhesiveness of the tip. This claim is not backed up experimentally and should therefore be toned down or avoided entirely.
7. The authors present low resolution data of co-purified filaments. The 2D classification of the wider filament looks as though there are issues with particle alignment. Did the authors try to add more particles or increase the box size to enhance the signal?
8. The suggestion that the 3 nm filament corresponds to B-DNA is not convincing at this resolution. It would be important to state why (if at all) a protein filament can be excluded and to tone down the assignment as B-DNA, unless further experimentation confirms it.
9. Please include the following standard cryoEM supplementary data: (i) A cryoEM data table, (ii) a cryoEM processing flowchart, (iii) a map / model FSC.
10. Lines 282 – 284: The claim "Interestingly, in *S. sanguinis* T4P, flexibility in the pilin subunits appears to be restricted to the melted region, which might play a direct role in the inherent flexibility of the filaments" would need to be supported experimentally, e.g. by 3D flexibility analysis

Minor comments:

1. Arrows or arrowheads indicating pili should be added to figure 1.
2. Line 68: Can a model organism be cutting-edge?
3. Line 90: EM not defined. Use electron microscopy.
4. Line 170: How well did the AF model fit in the map?
5. Include an RMSD plot to support the statement in line 190
6. Line 251: Delete the word "strongly".
7. Line 257 – 258: Replace "It shows how minor pilins cap the pilus tip efficiently" with "It proposes how minor pilins may cap the pilus"?
8. Line 258: Delete "tip efficiently".
9. Line 270: Replace "shines" with "sheds".
10. Line 330: Replace: "it is not an experimentally determined structure, and it is thus perfectible" with "Is merely a prediction and therefore requires experimental validation".
11. There is some self-gratifying language in places, which should be avoided. E.g. in line 272 "An important achievement in this study is that"; line 326 "major achievement" and line 327 "we are in a privileged position".

12. Line 315 & 316: The words "highly" are unnecessarily hyperbolic. Delete.
13. Lines 322 – 323: Clarify if the mutants move slower or faster.
14. Line 324: Replace "and it is thus perfectible" with "and therefore requires experimental validation."
15. Line 345 – 346: Replace "electron cryotomography improvements" with "new electron tomography tools".

Reviewer #1

1. Line 94: “predominantly polar”. How can they possibly tell from low resolution negative stain images? Most helical polymers (such as F-actin, microtubules, etc.) are polar, meaning that the two ends are different and all subunits have the same orientation in the filament. In contrast, there are some bi-polar polymers (such as muscle thick filaments) where the two ends are identical. But for many of these structures polarity is only evident at high resolution. For F-actin, polarity can be visualized at lower resolution using decoration by myosin S1, generating very polar arrowheads.

This was not properly phrased and has thus been reworded. What we meant is that pili emanate most often from the old poles of *S. sanguinis* diplococci (opposite from the division site). This can be seen in Fig. 1.

2. Line 94-5: “6.78 nm in diameter”. How can the diameter be determined to a precision of a tenth of an Å from low resolution negative stain images? This is meaningless, and it is better to say ~ 7 nm in diameter for these dried and presumably partially flattened filaments.

This was an average value from many measurements. However, we agree with the comment and have changed the text as suggested.

3. Lines 104-106: No details have been provided about the image processing of the thick filaments and the reconstruction in Fig. S1B. The paper states that 393,556 particles were used, but this is presumably for the 7 nm filaments. How many were used for the thick filaments?

How the images were processed – including for the thick filaments – is now better described in the *Image processing* section of the Materials and methods, which has been extensively modified. In addition, we have schematized the cryo-EM data processing workflow in Fig. S2.

4. Line 106-7: It is suggested that the thick filaments may result from a dramatic change in quaternary structure “similarly to what has been reported for *N. gonorrhoeae* T4P” [ref. 26]. But in ref. 26, it is shown that mechanical stretching of the T4P results in much thinner filaments, the exact opposite of what is being described here!

The adverb “similarly”, which only meant that gonococcal T4P exhibited a quaternary change too, was misleading. This has been rephrased.

5. Lines 110-113: The observation of extracellular DNA is very nice, but the authors may be unaware of the extensive discussion of this in a related area, T4P from *Geobacter sulfurreducens*. Many papers, primarily from Lovley and colleagues, have argued that “3 nm” extracellular filaments seen by AFM are T4P formed from only the PilA-N chain, and lacking a globular C-terminal domain. But other labs using cryo-EM have shown that these 3 nm filaments are DNA which is abundant in many preparations (Gu et al., 2021; Wang et al., 2022). This issue is discussed in a recent review (Wang et al., 2023).

These papers have now been referenced in the manuscript.

6. Lines 125-126: There is no explanation of exactly what was done for the imposition of helical symmetry (Fig. S2). Was this done with the 393,556 particles? Were the helical parameters allowed to evolve?

How this was done has now been explained in the *Image processing* section in the Materials and methods.

7. Lines 160-164: It is argued that PilE1 and PilE2 are distributed randomly in the filament. It is noted that in an archaeal flagellar filament, two isoforms of the same protein are arranged into hexamers, and not randomly. A very simple observation, the power spectrum from the filament segments, shows that this is the case in the archaeal flagellar filaments, where it is easily seen that the asymmetric unit is a hexamer with six times the axial rise expected for an asymmetric unit containing only one protein subunit. The authors should provide a power spectrum here showing that there is indeed no higher order symmetry arising from a regular arrangement of PilE1 and PilE2.

Since *S. sanguinis* T4P is bent, unlike the heteropolymeric archaeal flagellum from *M. villosus*, averaged power spectra are of very poor quality and cannot be used to assess higher order symmetry. This is one of the main reasons we chose a processing strategy with no symmetry imposed.

8. Line 202: “velcro” needs to be capitalized, as it is a brand name.

This has been done.

9. Lines 229-230: *"We previously determined the structure and function of the three other subunits of S. sanguinis T4P, the minor pilins PilA, PilB and PilC". But these structures were not of the whole pilins, only the globular domains.*

This has been spelled out clearly.

10. Line 256: *"at atomic resolution". This structure is not at atomic resolution (~ 1.2 Å). Rather, it is at "near-atomic resolution".*

This has been corrected.

11. Line 257: *"our model...is invaluable". This is a level of hyperbole that does not belong in the paper. Further, the model for how the minor pilins cap the filament is only a model, and not an experimental structure.*

This has been rephrased as suggested.

12. Line 271: *"notable findings" is again hyperbole that could be reduced to "findings".*

This has been rephrased as suggested.

13. Line 331: *Saying that they do not have an experimental structure and their model "is thus perfectible" implies that a true experimental structure is perfect. It would be much better to simply say that their model needs to be proved or improved experimentally.*

This has been rephrased as suggested.

14. Line 358: *"AlphaFold model of PilAC, which confirmed the interaction interface characterised experimentally" is not the best phrasing. I do not think that an AlphaFold model can confirm anything. For example, just several sentences later it is said that "This led to significant remodelling of the interface between PilB and PilC proposed by AlphaFold in the PilABC complex." Rather, one might say that the AlphaFold model was consistent with an interaction interface characterized experimentally.*

This has been rephrased as suggested.

15. Lines 370-1: *"melting in α1N of pilins occurring during filament assembly, which similarly could not be predicted". AlphaFold predictions are not mystical nor are they based upon physical principles. Rather, they are simply machine learning from a large training set (the PDB). As more structures of T4P with partially melted helices, such as this one, are deposited in the PDB, such melting will be "predicted" in the future.*

We agree with this point and have rephrased that sentence.

16. Lines 373-4: *"solving the structure of a T4F in a bacterial species radically different from previously characterized ones" is probably a poor choice of words. I would phrase it simply as a monoderm, whereas all previous T4F structures have been from Gram-negative bacteria.*

This has been rephrased as suggested.

17. Line 458: *"a separate distance between segments of 0.7 and a minimum filament length to consider of 1." This may have meaning to people who use cryoSPARC but what is a distance of 0.7 and a length of 1?*

We agree that this was not clearly phrased. Therefore, as mentioned above, this section of the Material and methods has been extensively rewritten, and all the distances are now indicated in Å.

Reviewer #2

1. *The authors refer to type 4 filaments as nanomachines. To this, I have more of a semantic question: It would seem that T4P per se are not nanomachines but part of a nanomachine that includes all the other proteins of the T4P machine to function. So maybe consider rewriting that T4F are the filamentous parts of a large nanomachine.*

This is an interesting point. We agree that the filament *per se* is just a part of the nanomachine, but it constitutes its business end, arguably the most important component. However, while it might be semantically more accurate to refer to them as "nanomachines centred on T4F", this

is in our view unnecessarily cumbersome. In the sake of simplicity, although it might be a "semantic shortcut", we refer to the entire nanomachine as T4F (a term now widely used). However, to address the comment of the reviewer, we have modified the beginning of the Introduction.

2. Line 45: Maybe rewrite to "structure of mature pilins".

This has been changed to "structure of full-length pilins".

3. Line 48: Maybe rewrite to "packed against the globular".

This has been changed to "part of the globular head".

4. Line 96: Please include arrow(s) in Fig. 1 to indicate the T4P that "aggregate laterally".

Arrowheads have been added in Fig. 1 to indicate individual filaments. It can be readily seen that a few filaments emanate from the same cellular location, aggregating laterally at their base. This has been clearly spelled out.

5. Fig. 3B: It is difficult to distinguish the two different subunits. Maybe to show the two different subunits, color them differently (e.g., with shades of cyan, purple, grey and green). Also, please include both pilins in Fig. 3B. Fig. S3A-C very nicely helps to show the differences between the two pilins in the solved structure. Because a major point of the manuscript is that the *S. sanguinis* T4P is composed of two major pilins, I suggest moving Fig. S3A-C to a figure in the main text.

There appears to be some misunderstanding here. Although individual pilin subunits can be visualised in Fig. 3B (one subunit has been highlighted in bold to make this point clear), we cannot distinguish between the PilE1 and PilE2 subunits. Indeed, the regions of difference between PilE1 and PilE2 – primarily the 8-aa loop in PilE1 – correspond to areas of significantly lower quality in the map and could not be modelled at all. This has been further explained in the text. To help the reader understand this critical point in the manuscript, we have extensively modified Fig. 3 by combining it with what was previously Fig. S3, as suggested by the Reviewer.

6. Line 155-164: Please include the control experiments in which the four PilE1 and PilE2 constructs are tested separately in the bacterial two-hybrid analyses. These controls are essential to ensure that none of the four constructs self-activate. Concerning the comment that the beta-gal level reflects the strength of binary interactions, this is not entirely correct. This level also depends on the accumulation level of the different fusion proteins. So, if statements are made about the "strength of interactions", then western blots should be included to demonstrate the level of accumulation of the four proteins.

In the BACTH system that we used, proteins of interest A and B are genetically fused to the T25 and T18 fragments that constitute the catalytic domain of *B. pertussis* adenylate cyclase. Since T25 and T18 cannot interact by themselves, there is therefore no possibility for "self-activation" as raised by the Reviewer. The only way there can be functional complementation between T25 and T18, which would lead to cAMP synthesis, is if proteins A and B interact (Karimova *et al.* 1998). cAMP then triggers transcriptional activation of sugar catabolic operons, which yields readily assayable and quantifiable phenotypes in an *E. coli* *cya* mutant. The suitable negative and positive controls – always the same in all BACTH papers – are therefore empty T15 and T18 plasmids, and plasmids in which these domains were fused to a yeast leucine zipper known to mediate strong interaction. These controls were already included in the paper. Although this is beyond the scope of this paper, we have recently extended the BACTH analysis to all the pilins, which showed that most of the pairs that were tested did not show any functional complementation, in contrast to the results reported here for PilE1 and PilE2.

As for the second point, quantification of the β -gal activities in liquid measures the efficiency of complementation between T25 and T18 fusions (Karimova *et al.* 1998). Since PilE1 and PilE2 are almost identical in sequence, and we have previously shown that they are equally well-behaved proteins when expressed in *E. coli* (Berry *et al.* 2019), it seems fair to consider that similar β -gal activities reflect similar affinities. This has been clarified in the text.

7. Line 175-191: I have several questions/comments about this para and the relevant figures:

(i) In Fig. 3 and S3A-C the solved structures of PilE1 and PilE2 obtained from the pili are shown. Why then are AlphaFold models of PilE1 and PilE2 used to generate the atomic model? If there is a good explanation, please include it.

(ii) in the legend to Fig. 5, please include whether the structures shown are solved structures or AlphaFold models. Also, the globular domains are described as having a three-stranded beta-sheet flanked at the N- and C termini with regions "with no distinctive secondary structure". However, in Fig. 3 and S3, these regions are described as alpha-helical. Please clarify.

(iii) Please include a more detailed explanation of what is shown in Fig. S4.

(i) Again, as in point 5 above, it is important to stress out that we could not distinguish between PilE1 and PilE2 in the density map, which is the reason we used the PilE1 AlphaFold model to generate the atomic model of the pilus. This has been spelled out.

(ii) This has been clarified.

(iii) A detailed explanation is provided.

8. For all AlphaFold models, please include the standard quality checks, including pAE and pLDDT diagrams of five models. Please also describe which model was picked for further analysis.

The standard quality checks have been included for all the models. We always used the models with the best statistics.

Reviewer #3

Major comments

1. The authors base their hypothesis that the filament is a random mosaic purely on the fact that an 8 aa loop between residues 92 and 100 is not as well resolved as the rest of the protein. While the hypothesis is intriguing, further evidence should be provided to substantiate it. NMR studies suggesting low flexibility of this area are not sufficiently convincing. Without further experimental support, the interpretation of the map and all derived discussion should be toned down.

The hypothesis of a mosaic pilus is supported by two lines of experimental evidence, not just the observation that the 8-aa loop present in PilE1 and absent in PilE2 is not well resolved in the cryo-EM structure. Firstly, this poor resolution cannot be explained by an inherent flexibility of the 8-aa loop since our previously reported NMR structure of PilE1 (Berry *et al.* 2019) showed no significant flexibility in this region. NMR spectroscopy is a reliable and widely accepted source of information about protein flexibility. Secondly, we show here by performing a BACTH analysis that PilE1 and PilE2 interact equally well with themselves as with one another, which argues against the possibility that *S. sanguinis* T4P are heteropolymers in which the two major subunits would alternate regularly upon polymerisation of pre-formed PilE1PilE2 heterodimers. Such a scenario has been recently described for the archaeum of *M. villosus*, the only other heteropolymeric T4F that has been structurally characterised in which the two subunits alternate regularly. However, we do agree that *S. sanguinis* pilus mosaicism remains hypothetical, the corresponding parts of the text have been toned down a suggested.

2. There is a lot of emphasis on the 8 aa loop between residues 92 and 100. Could the authors provide a figure panel (at least in the supplementary figures) to show this lower resolved area in comparison with the rest of the high-resolution map?

This was already shown in Fig. S3, which has been included in Fig. 3. To make it easier to understand, we have highlighted the 8-aa loop in Fig. 3.

3. The model of the tip is very interesting. However, it is mostly based on an AlphaFold prediction and would thus require experimental verification before substantial functional conclusions are drawn. I appreciate that this would be technically challenging, so therefore, the interpretation of the model needs to be toned down.

The model of the tip is supported by experimental evidence published in a recent paper (Shahin *et al.* 2023). This has been clarified in the manuscript. However, since this remains a model, the corresponding parts of the text have been toned down a suggested.

4. Furthermore, the AlphaFold model needs to be corroborated by confidence plots.

Values for the standard AlphaFold quality checks have been included for all the models.

5. The model of the tip suggests that the hydrophobic alpha helices of PilC and PilA are exposed to the surrounding medium (Fig. 7), which does not seem very convincing. The authors should critically assess this or provide an explanation.

This is a misinterpretation of the Fig.7. The α 1-helices of PilA and PilC align very well with the α 1-helices of major subunits at the apex of T4P. This ensures that the helical symmetry of the pilus is maintained, and that the hydrophobic α 1N portions of PilA and PilC are buried deeply in the filament core (not exposed to the surrounding medium as mentioned by the Reviewer). This can be easily seen when the major pilin subunits are made transparent.

6. It is suggested that the “wings” of the tip flap about and that this enhances the adhesiveness of the tip. This claim is not backed up experimentally and should therefore be toned down or avoided entirely.

Again, this flexibility has been proposed in a recent paper (Shahin *et al.* 2023). However, since this remains hypothetical, the corresponding parts of the text have been toned down as suggested.

7. The authors present low resolution data of co-purified filaments. The 2D classification of the wider filament looks as though there are issues with particle alignment. Did the authors try to add more particles or increase the box size to enhance the signal?

Although we have performed new rounds of image processing – by notably trying different box sizes and even attempting 3D classifications – we have not been able to improve the 2D averages and reconstructions for the thick filaments. It is likely that these filaments are very heterogeneous in their conformation and/or flexible, which prevents correct alignment during 2D/3D classification and reconstruction.

8. The suggestion that the 3 nm filament corresponds to B-DNA is not convincing at this resolution. It would be important to state why (if at all) a protein filament can be excluded and to tone down the assignment as B-DNA, unless further experimentation confirms it.

As mentioned in response to Reviewer #1 (see point 5), ours is not the only cryo-EM study identifying 3 nm filaments, which are abundant in many pilus preparations, as DNA. This has now been discussed in the manuscript and the corresponding papers have been cited (Gu *et al.* 2021; Wang *et al.* 2022; Wang *et al.* 2023).

9. Please include the following standard cryo-EM supplementary data: (i) A cryo-EM data table, (ii) a cryo-EM processing flowchart, (iii) a map / model FSC.

All the requested data have been included in the revised manuscript. (i) Table S1 list all the cryo-EM data collected and the refinement statistics. (ii) Fig. S2 presents the cryo-EM processing flowchart that was used in this study. (iii) Fig. S3B presents the map versus model FSC curves, with and without mask, for the final model.

10. Lines 282 – 284: The claim “Interestingly, in *S. sanguinis* T4P, flexibility in the pilin subunits appears to be restricted to the melted region, which might play a direct role in the inherent flexibility of the filaments” would need to be supported experimentally, e.g., by 3D flexibility analysis.

This has been rephrased and tone down.

Minor comments

1. Arrows or arrowheads indicating pili should be added to figure 1.

This has been added.

2. Line 68: Can a model organism be cutting-edge?

This has been reworded.

3. Line 90: EM not defined. Use electron microscopy.

This has been done.

4. Line 170: How well did the AF model fit in the map?

Details are provided in the text. Fig. 5A illustrates how well the N-ter peptide backbone of the pilin – corresponding to the α 1 helix – fits within our final cryo-EM density map. The structure of the subunit in the filament – shown in Fig. 5B – aligns perfectly well with the AlphaFold PilE1 model with a mere 1.37 Å RMSD.

5. Include an RMSD plot to support the statement in line 190

The RMSD value mentioned (1.3 Å) indicates how well the structures of fully resolved subunits in our pilus structure superpose. As far as we know, RMSD plots are used in molecular dynamics simulations to show the deviation that occurs during the period of stimulation, not in cryo-EM studies such as ours.

6. Line 251: Delete the word “strongly”.

This has been done.

7. Line 257 – 258: Replace “It shows how minor pilins cap the pilus tip efficiently” with “It proposes how minor pilins may cap the pilus”?

This has been done.

8. Line 258: Delete “tip efficiently”.

This has been done.

9. Line 270: Replace “shines” with “sheds”.

This has been done.

10. Line 330: Replace: “it is not an experimentally determined structure, and it is thus perfectible” with “Is merely a prediction and therefore requires experimental validation”.

This has been done.

11. There is some self-gratifying language in places, which should be avoided. E.g., in line 272 “An important achievement in this study is that”; line 326 “major achievement” and line 327 “we are in a privileged position”.

This has been done.

12. Line 315 & 316: The words “highly” are unnecessarily hyperbolic. Delete.

This has been done.

13. Lines 322 – 323: Clarify if the mutants move slower or faster.

This has been done.

14. Line 324: Replace “and it is thus perfectible” with “and therefore requires experimental validation.”

This has been done.

15. Line 345 – 346: Replace “electron cryotomography improvements” with “new electron tomography tools”.

This has been done.

Reviewers' Comments:

Reviewer #1:

Remarks to the Author:

The authors have done a very thorough job of addressing my concerns. The paper makes a significant contribution to our understanding of type IV pili and should now be published.

Edward H. Egelman

Reviewer #2:

Remarks to the Author:

The authors have very nicely addressed all my comments.

Congratulations on a very exciting piece of work.

Reviewer #3:

Remarks to the Author:

The authors have addressed all my comments.

Reviewer #1

The authors have done a very thorough job of addressing my concerns. The paper makes a significant contribution to our understanding of type IV pili and should now be published.

Edward H. Egelman

Reviewer #2

The authors have very nicely addressed all my comments. Congratulations on a very exciting piece of work.

Reviewer #3

The authors have addressed all my comments.

We have addressed all the editorial requests as you will see in the final version of our manuscript and in the attached documents. We have edited our manuscript to comply with your policies and formatting requirements. All the changes are highlighted in the text.